

# Losartan and dapagliflozin combination therapy in reducing uric acid level compared to monotherapy in patients with heart failure

Tuong Le Trong Huynh[1], Phong Thanh Pham[1], Hien Dieu Tran[1], Nhan Dinh Tran[1], Duong Van Tran[1], Bao Lam Thai Tran[2], Khoa Dang Dang Tran[2], Toan Hoang Ngo[2] and Son Kim Tran[2]

[1] Department of Cardiology and Rheumatology, Can Tho Central General Hospital, Can Tho, Ninh Kieu, Vietnam
[2] Department of Internal Medicine, Can Tho University of Medicine and Pharmacy, Can Tho, Ninh Kieu, Vietnam

Corresponding author
Toan Hoang Ngo,
nhtoan@ctump.edu.vn

## ABSTRACT

**Background:** Sodium-Glucose Transport Protein 2 (SGLT2) inhibitors, and Angiotensin II Receptor Antagonists (ARBs) also have the effect of reducing serum uric acid but few studies worldwide assessed.
**Objective:** Evaluate the effectiveness of serum uric acid lowering treatment of SGLT2 inhibitors, and ARB in heart failure (HF) patients.
**Methods:** We conducted a cross-sectional analysis study with 8 weeks of follow-up on 733 heart failure (HF) patients treated at Can Tho Central General Hospital from January 2023 to March 2024. Patients enrolled in the study were examined and received losartan (Group A) or dapagliflozin (Group B) monotherapy or losartan and dapagliflozin combined therapy (Group C). The uric acid concentration group was defined into three subgroups with tertile 1 from smallest to quartile (Q) 1, tertile 2 from Q2 to Q3, and tertile 3 from Q3 to the largest value.
**Results:** After 8 weeks of treatment, the uric acid reduction effect between groups A, B, and C showed that the combination group had the optimal reducing effect compared to losartan and dapagliflozin monotherapy with the mean difference being $-229.62 \pm 76.65$ μmol/L, $-217.00 \pm 146.17$ μmol/L, and $-284.43 \pm 136.32$ μmol/L, respectively. In total, combination therapy showed the best reduction outcome in the population of male, female, patients with type 2 diabetes mellitus (T2DM), and dyslipidemia with the mean difference ranging from $-226.21 \pm 74.65$ μmol/L to $-231.85 \pm 76.28$ μmol/L and $-209.62 \pm 184.94$ μmol/L to $-225.75 \pm 78.53$ μmol/L and $-273.02 \pm 204.54$ μmol/L to $-308.93 \pm 72.97$ μmol/L in group A, B, and C, respectively.
**Conclusion:** The optimal therapy for reducing uric acid levels in HF patients was the combination of losartan and dapagliflozin, and the effectiveness did not change through sex, T2DM, and dyslipidemia patients.

## INTRODUCTION

Heart failure (HF) is an increasingly common syndrome and a common problem worldwide. In the United States (US), according to statistics from the 2000s, there were 5 million HF patients, including about 550,000 new cases and 285,000 deaths annually. The rate of HF patients over 65 years old was 6.6–9.8% and the 5-year mortality rate in men was 59% and in women was 45%. It was estimated that by 2037 the number of heart failure patients could reach 10 million people, doubling the number of patients in 2000 in the next 40 years (*Lloyd-Jones et al., 2010*). In addition, HF could be a culprit for insulin resistance and greatly affects the patient's health status and quality of life (*Son et al., 2022*). According to the prevalence of the disease worldwide, it was estimated that about 320,000 to 1.6 million people in Vietnam have heart failure and most departments were overloaded with HF patients in the hospital. This is truly an economic burden for the family and the whole of society (*Vietnamese Ministry of Health, 2008*).

Heart failure is associated with several risk factors including hypertension, coronary artery disease, arrhythmia, and diabetes (*Chamberlain et al., 2020*). In addition, hyperuricemia and HF often occur in people with metabolic syndrome (*Dobrowolski et al., 2022*; *Jakubiak et al., 2024*). Ischemia will increase xanthine oxidation activity and uric acid synthesis. Therefore, increased uric acid can be a marker of myocardial ischemia. Increased uric acid has a predictive value for mortality as well as predicting the occurrence of cardiovascular events (CEs) in patients with HF or coronary artery disease (CAD) and a negative prognostic factor for moderate to severe HF (*Nadkar & Jain, 2008*). Additionally, chronic kidney disease (CKD) has an association with increasing the prevalence of hyperuricemia and mortality risk in patients with HF (*Wang et al., 2023a*). Besides the role of treating HF, drugs in the group of Sodium-Glucose Transport Protein 2 (SGLT2) inhibitors, Angiotensin-Converting Enzyme Inhibitors (ACEi), and Angiotensin II Receptor Antagonists (ARBs) also have the effect of reducing serum uric acid (*Doehner et al., 2022*; *Kim et al., 2020*; *Masajtis-Zagajewska, Majer & Nowicki, 2021*; *McDowell et al., 2022*; *Schmidt et al., 2001*). In Vietnam, few studies evaluated the role of SGLT2 inhibitors, ACEi, and ARB as serum uric acid-reducing factors in HF patients. Therefore, we conducted a study to evaluate the effectiveness of serum uric acid lowering treatment of SGLT2 inhibitors, and ARB in HF patients.

## MATERIALS AND METHODS

### Study design and population

We conducted a cross-sectional analysis study with 8 weeks of follow-up on 733 HF patients treated at Can Tho Central General Hospital from January 2023 to March 2024.

Inclusion criteria: Heart failure patients were diagnosed according to the European Society of Cardiology (ESC) 2021 guideline (*Bauersachs & Soltani, 2022*) with symptoms (dyspnea during exertion or rest, fatigue, drowsiness, ankle edema), signs (heart rate tachypnea, shortness of breath, rales at the base of the lungs, pleural effusion, distended jugular veins, hepatomegaly, peripheral edema) and signs of structural or functional abnormalities of the resting heart (cardiomegaly, gallop T3, heart murmur, abnormal

echocardiogram, blood tests with increased Brain natriuretic peptide (BNP) or N-terminal pro-brain natriuretic peptide (NT-proBNP)). Patients agreed to participate in the study.

Exclusion criteria: Patients with chronic kidney failure (CKD), patients using other uric acid-lowering drugs such as allopurinol, and febuxostate, patients who do not agree to participate in the study.

## Sample size

Based on the study of *Hamaguchi et al. (2011)* in the Japanese population, the prevalence of hyperuricemia in HF patients was 56% ($p = 0.56$), với $\alpha = 0.05$ with $\alpha = 0.05$ corresponding to $Z_{1-\frac{\alpha}{2}} = 1.96$, and $d = 0.05$. According to the one-proportion sample estimation formula, we estimated $n = 378$. Therefore, we need to collect more than 378 patients for the study to be representative of the study population. We conducted research on 733 patients with HF during the study period.

## Study variables and data collection methods

Patients enrolled in the study were examined and collected: sex, age, blood pressure, body mass index (BMI) were categorized following the Asia-Pacific classification (*Beghin et al., 1988*), waist circumference, heart failure degree were classified through the New York Heart Association (NYHA) (*Hunt, 2005*). Comorbidities include hypertension (based on ESC/ESH 2018, hypertension when systolic blood pressure (SBP) ≥140 mmHg and/or diastolic blood pressure (DBP) ≥90 mmHg at least two measurements or using antihypertensive (*Williams et al., 2018*)), type 2 diabetes mellitus (T2DM) (according to American Diabetes Association (ADA) recommendations 2020 (*American Diabetes Association, 2020*)), dyslipidemia (according to National Cholesterol Education Program Adult Treatment Panel III (NCEP-ATP III) (*National Cholesterol Education Program (NCEP) Expert Panel on Detection, Evaluation, and Treatment of High Blood Cholesterol in Adults (Adult Treatment Panel III), 2002*)). Medical history includes a history of coronary artery diseases (CADs) (include history of myocardial infarction or ischemic heart disease) and a history of cerebral infarction (CI). Bad habits include smoking (according to Community Intervention Trial (*Mazzone et al., 2001*)) and alcohol consumption (positive when >3 units for males and >2 units for females (*Wakefield & Brennan, 2018*)).

Uric acid was defined as high when ≥7 mg/dL (420 μmol/L) in males, and ≥6 mg/dL (360 μmol/L) in females (*Waring, Webb & Maxwell, 2000*). The uric acid concentration group was defined into three subgroups with tertile 1 from smallest to quartile (Q) 1, tertile 2 from Q2 to Q3, and tertile 3 from Q3 to the largest value (*Doehner et al., 2022*). Cholesterol, triglyceride, high-density lipoprotein cholesterol (HDL-c), and low-density lipoprotein cholesterol (LDL-c) were collected and defined according to NCEP-ATP III (*National Cholesterol Education Program (NCEP) Expert Panel on Detection, Evaluation, and Treatment of High Blood Cholesterol in Adults (Adult Treatment Panel III), 2002*). Serum glucose and hemoglobin A1c (HbA1c) was also collected to control patients with and without T2DM. Doppler ultrasound in Adaptive Simpson's method (*Porter et al., 2018*) for measuring atrial and ventricular function. Ejection fraction (EF) to the

classification of heart failure (HF) as Heart failure with reduced ejection fraction (HFrEF) or heart failure with preserved ejection fraction (HFpEF).

Enrolled HF patients received losartan (Group A) or dapagliflozin (Group B) monotherapy or losartan and dapagliflozin combined therapy (Group C) based on the clinical requirements (Based on guidelines for HF management of ESC (*Bauersachs & Soltani, 2022*)). The study researcher who is a clinical physician specialist in cardiology indicated medical therapy for the enrolled patients. The initial treatment dose of losartan (Cozaar tablets containing 50 mg from Merck Sharp & Dohme, Rahway, NJ, USA) is 50 mg daily, then the dose can be increased to 150 mg daily. Dapagliflozin (Forxiga tablets containing 10 mg from AstraZeneca, Cambridge, UK) used 10 mg daily. All of the patients enrolled in the study were followed up to 8 weeks to assess the treatment outcome which was reduced uric acid concentration.

## Data bias control methods and data analysis

Physicians were trained to unify variables in data collection forms and procedures in the study. All of the machines were standardized before data collection. Tests are inspected internally and externally to meet requirements. Data entry into the software was done by two researchers.

Data were collected and processed by SPSS 25.0 software (IBM, Armonk, NY, USA). Quantitative variables with normal distribution were described by mean ± standard deviation (SD), and non-normal distribution variables were described by median, and interquartile range (IQR). Qualitative variables are described by proportion and percentage. The difference between two qualitative variables described by the Chi-squared test, normal distribution quantitative variables by simple t-test (if two groups analyzed) or ANOVA (if ≥3 groups analyzed), quantitative variables with non-normal distribution by Mann–Whitney test (if two groups analyzed) or Kruskal–Wallis test (if ≥3 groups analyzed). Multivariable analysis was used to assess related factors to uric acid treatment outcomes through Odds Ratio (OR), and $p < 0.05$ considered to be statistically significant.

## Ethical approval

The study was approved by the Ethics Committee in Biomedical Research of Can Tho University of Medicine and Pharmacy with Decision No. 182/HDDD-DHYDCT at December 22, 2022. All participating patients were asked to fill out a consent form to participate in the study. The identities of all patients were kept confidential.

## RESULTS

### Baseline characteristics of the study population

There were 733 HF patients enrolled in the analysis phase of our study, in which 246 patients received losartan, 247 patients received dapagliflozin and 240 patients received the combination of two drugs (Fig. 1). Table 1 showed the baseline characteristics of our study population within three groups of treatment. There was no difference in sex and age between the three groups. BMI, WC, EF, and NYHA grades were different between the three groups. Comorbidities with the highest prevalence were dyslipidemia and

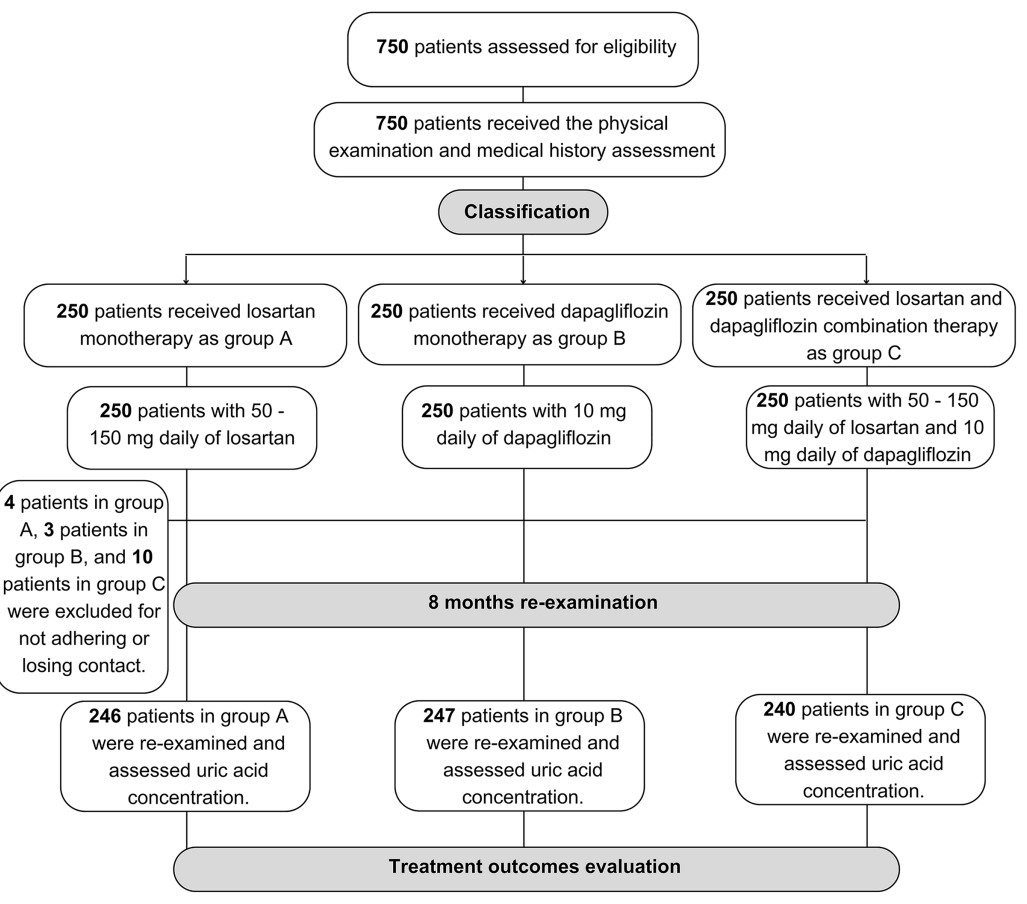

**Figure 1** **Sample collection flowchart of the study population.**

**Table 1 Baseline characteristics of the study population.**

| Characteristics | | Group A (*n* = 246) | | Group B (*n* = 247) | | Group C (*n* = 240) | | $p^a$ |
|---|---|---|---|---|---|---|---|---|
| | | *n* | % | *n* | % | *n* | % | |
| Male | | 125 | 50.8 | 113 | 45.7 | 112 | 46.7 | 0.488 |
| Age (years), mean ± SD | | 68.50 ± 12.48 | | 67.34 ± 13.54 | | 68.06 ± 14.59 | | 0.631[b] |
| Blood pressure (mmHg), mean ± SD | Systolic | 125.49 ± 21.77 | | 123.91 ± 22.79 | | 120.04 ± 16.97 | | 0.012[b] |
| | Diastolic | 76.83 ± 19.95 | | 75.58 ± 14.21 | | 73.60 ± 11.17 | | 0.025[b] |
| BMI (kg/m$^2$), mean ± SD | | 20.95 ± 3.11 | | 20.90 ± 3.07 | | 20.18 ± 2.54 | | 0.005[b] |
| WC (cm), mean ± SD | | 79.00 ± 9.16 | | 78.69 ± 8.55 | | 76.65 ± 6.77 | | 0.003[b] |
| NYHA grade | II | 128 | 52.0 | 128 | 51.8 | 51 | 21.3 | <0.001 |
| | III | 112 | 45.5 | 112 | 45.3 | 174 | 72.5 | |
| | IV | 6 | 2.4 | 7 | 2.8 | 15 | 6.3 | |
| Hypertension | | 205 | 83.3 | 208 | 84.2 | 222 | 92.5 | 0.005 |
| T2DM | | 105 | 42.7 | 108 | 43.7 | 129 | 53.8 | 0.026 |
| Dyslipidemia | | 219 | 89.0 | 224 | 90.7 | 219 | 91.3 | 0.688 |

(Continued)

| Table 1 (continued) | | | | | | | |
| Characteristics | | Group A (n = 246) | | Group B (n = 247) | | Group C (n = 240) | $p^a$ |
| | | n | % | n | % | n | % | |
| History of CAD | | 127 | 51.6 | 133 | 53.8 | 124 | 51.7 | 0.853 |
| History of CI | | 83 | 33.7 | 78 | 31.6 | 94 | 39.2 | 0.195 |
| Smoking | | 94 | 38.2 | 77 | 31.2 | 95 | 39.6 | 0.116 |
| Alcohol consumption | | 102 | 41.5 | 97 | 39.7 | 98 | 40.8 | 0.878 |
| EF (%), mean ± SD | | 34.73 ± 8.16 | | 35.61 ± 7.99 | | 36.70 ± 9.14 | | $0.037^b$ |
| Uric acid (µmol/L) | Tertile 1 | 72 | 29.3 | 66 | 26.7 | 45 | 18.8 | <0.001 |
| | Tertile 2 | 134 | 54.5 | 131 | 53.0 | 102 | 42.5 | |
| | Tertile 3 | 40 | 16.3 | 50 | 20.0 | 93 | 38.8 | |
| | Mean ± SD | 495.41 ± 109.39 | | 504.56 ± 111.80 | | 548.78 ± 123.49 | | $<0.001^b$ |
| Cholesterol (mmol/L), median (IQR) | | 4.00 (3.10–4.70) | | 4.00 (3.30–4.80) | | 4.10 (3.40–4.70) | | $0.595^c$ |
| Triglyceride (mmol/L), median (IQR) | | 1.2 (0.93–1.90) | | 1.20 (1.00–1.90) | | 1.30 (1.00–1.90) | | $0.725^c$ |
| HDL-c (mmol/L), median (IQR) | | 1.00 (0.80–1.20) | | 1.00 (0.80–1.10) | | 0.90 (0.80–1.10) | | $0.886^c$ |
| LDL-c (mmol/L), median (IQR) | | 2.55 (2.00–3.10) | | 2.60 (2.00–3.10) | | 2.70 (2.10–3.17) | | $0.414^c$ |

Notes:
CAD, coronary artery disease; CI, cerebral infarction; EF, ejection fraction; WC, waist circumference; NYHA, The New York Heart Association; SD, standard deviation; HDL-c, high density of lipoprotein cholesterol; LDL-c, low density of lipoprotein cholesterol; IQR, interquartile range.
[a] Chi-squared test.
[b] One-way ANOVA test.
[c] Kruskal–Wallis U test.

hypertension. The mean concentration of uric acid at baseline of groups A, B, and C was 495.41 ± 109.39 µmol/L, 504.56 ± 111.80 µmol/L, and 548.78 ± 123.49 µmol/L, respectively. Uric acid concentration within three tertiles between three groups showed the difference. Bilan lipid was not different within groups of treatment (Table 1).

## Treatment outcome

After 8 weeks of treatment, the uric acid reduction effect between groups A, B, and C showed that the combination group had the optimal reducing effect compared to losartan and dapagliflozin monotherapy with the mean difference was −229.62 ± 76.65 µmol/L, −217.00 ± 146.17 µmol/L, and −284.43 ± 136.32 µmol/L, respectively (Table 2). Subgroup analysis between the tertile of the uric acid level also indicated the optimal effect in the group with combination therapy. The effect was shown most clearly in tertiles 2 and 3 with the mean difference of groups A, B, and C being −248.63 ± 61.15 µmol/L, −221.38 ± 182.97 µmol/L, −303.67 ± 50.29 µmol/L respectively in tertile 2, and −292.90 ± 62.98 µmol/L, −289.04 ± 57.06 µmol/L, −354.61 ± 48.52 µmol/L respectively in tertile 3 ($p < 0.05$) (Table 2).

In addition, our study also assessed the uric acid level reduction effectiveness of the subgroup population for such as sex, T2DM, and dyslipidemia patients. In total, combination therapy showed the best reduction outcome in the population of male, female, patients with T2DM, and dyslipidemia with the mean difference ranging from −226.21 ± 74.65 µmol/L to −231.85 ± 76.28 µmol/L and −209.62 ± 184.94 µmol/L to

**Table 2 Uric acid reduction outcome within groups of treatment.**

| Uric acid (µmol/L) | | Group A (n = 246) | | Group B (n = 247) | | Group C (n = 240) | | $p^{a}$ |
|---|---|---|---|---|---|---|---|---|
| | | n | % | n | % | n | % | |
| Total | Endpoint archive | 227 | 92.3 | 223 | 90.3 | 235 | 97.9 | 0.002 |
| | Mean ± SD | 265.79 ± 74.81 | | 287.56 ± 151.19 | | 264.35 ± 139.74 | | 0.075[b] |
| | Mean difference | −229.62 ± 76.65 | | −217.00 ± 146.17 | | −284.43 ± 136.32 | | <0.001[b] |
| Tertile 1 (n = 183) | Endpoint archive | 72 | 100.0 | 66 | 100.0 | 44 | 97.8 | 0.214 |
| | Mean ± SD | 206.99 ± 41.62 | | 205.26 ± 38.67 | | 238.67 ± 300.47 | | 0.464[b] |
| | Mean difference | −159.07 ± 55.72 | | −153.72 ± 57.84 | | −95.75 ± 290.59 | | 0.068[b] |
| Tertile 2 (n = 367) | Endpoint archive | 129 | 96.3 | 120 | 91.6 | 102 | 100.0 | 0.007 |
| | Mean ± SD | 267.65 ± 61.18 | | 301.50 ± 189.14 | | 237.44 ± 47.52 | | <0.001[b] |
| | Mean difference | −248.63 ± 61.15 | | −221.38 ± 182.97 | | −303.67 ± 50.29 | | <0.001[b] |
| Tertile 3 (n = 183) | Endpoint archive | 26 | 65.0 | 37 | 74.0 | 89 | 95.7 | <0.001 |
| | Mean ± SD | 365.40 ± 51.62 | | 359.68 ± 54.01 | | 306.28 ± 46.59 | | <0.001[b] |
| | Mean difference | −292.90 ± 62.98 | | −289.04 ± 57.06 | | −354.61 ± 48.52 | | <0.001[b] |

Notes:
SD, standard deviation.
[a] Chi-squared test.
[b] One-way ANOVA test.

**Table 3 Uric acid reduction outcome in subgroups within groups of treatment.**

| Uric acid (µmol/L) | | Group A (n = 246) | | Group B (n = 247) | | Group C (n = 240) | | $p^{a}$ |
|---|---|---|---|---|---|---|---|---|
| | | n | % | n | % | n | % | |
| Total | | | | | | | | |
| Male (n = 350) | Endpoint archive | 121 | 96.8 | 109 | 96.5 | 111 | 99.1 | 0.391 |
| | Mean ± SD | 262.15 ± 76.09 | | 281.00 ± 76.94 | | 257.70 ± 59.36 | | 0.034[b] |
| | Mean difference | −226.21 ± 74.65 | | −225.75 ± 78.53 | | −297.46 ± 91.84 | | <0.001[b] |
| Female (n = 383) | Endpoint archive | 106 | 87.6 | 114 | 85.1 | 124 | 96.9 | 0.004 |
| | Mean ± SD | 269.55 ± 73.58 | | 293.09 ± 192.94 | | 170.16 ± 183.29 | | 0.405[b] |
| | Mean difference | −233.13 ± 78.82 | | −209.62 ± 184.94 | | −273.02 ± 204.54 | | 0.009[b] |
| T2DM (n = 342) | Endpoint archive | 95 | 90.5 | 96 | 88.9 | 125 | 96.9 | 0.046 |
| | Mean ± SD | 276.56 ± 75.28 | | 302.24 ± 212.92 | | 266.64 ± 63.49 | | 0.112[b] |
| | Mean difference | −231.85 ± 76.28 | | −204.32 ± 204.34 | | −308.93 ± 72.97 | | <0.001[b] |
| Dyslipidemia (n = 662) | Endpoint archive | 204 | 93.2 | 203 | 90.6 | 214 | 97.7 | 0.007 |
| | Mean ± SD | 264.41 ± 73.16 | | 286.37 ± 156.69 | | 264.24 ± 145.41 | | 0.112[b] |
| | Mean difference | −229.84 ± 77.44 | | −213.04 ± 151.35 | | −280.53 ± 167.99 | | <0.001[b] |
| Tertile 1 (n = 183) | | | | | | | | |
| Male (n = 87) | Endpoint archive | 37 | 100.0 | 32 | 100.0 | 18 | 100.0 | – |
| | Mean ± SD | 207.30 ± 44.23 | | 201.59 ± 35.58 | | 189.94 ± 17.81 | | 0.272[b] |
| | Mean difference | −164.45 ± 56.99 | | −154.00 ± 62.03 | | −126.44 ± 43.47 | | 0.070[b] |
| Female (n = 96) | Endpoint archive | 35 | 100.0 | 34 | 100.0 | 29 | 96.3 | 0.275 |
| | Mean ± SD | 206.66 ± 39.31 | | 208.71 ± 41.36 | | 271.15 ± 387.09 | | 0.407[b] |
| | Mean difference | −153.37 ± 54.59 | | −153.47 ± 54.54 | | −75.29 ± 374.94 | | 0.245[b] |

(Continued)

| Table 3 (continued) | | | | | | | | |
|---|---|---|---|---|---|---|---|---|
| Uric acid (µmol/L) | | Group A (n = 246) | | Group B (n = 247) | | Group C (n = 240) | | $p^a$ |
| | | n | % | n | % | n | % | |
| T2DM (n = 61) | Endpoint archive | 21 | 100.0 | 27 | 100.0 | 13 | 100.0 | – |
| | Mean ± SD | 215.48 ± 40.66 | | 206.96 ± 47.17 | | 198.62 ± 30.13 | | 0.516[b] |
| | Mean difference | −148.00 ± 45.22 | | −152.26 ± 63.09 | | −161.84 ± 27.62 | | 0.747[b] |
| Dyslipidemia (n = 169) | Endpoint archive | 63 | 100.0 | 62 | 100.0 | 43 | 97.7 | 0.240 |
| | Mean ± SD | 208.17 ± 41.32 | | 205.24 ± 35.54 | | 240.25 ± 303.75 | | 0.480[b] |
| | Mean difference | −157.08 ± 54.74 | | −151.79 ± 55.27 | | −95.00 ± 293.90 | | 0.098[b] |
| Tertile 2 (n = 367) | | | | | | | | |
| Male (n = 173) | Endpoint archive | 71 | 100.0 | 50 | 96.2 | 50 | 100.0 | 0.095 |
| | Mean ± SD | 263.96 ± 61.31 | | 291.27 ± 61.89 | | 239.88 ± 42.20 | | <0.001[b] |
| | Mean difference | −244.45 ± 62.31 | | −233.42 ± 60.38 | | −306.78 ± 45.72 | | <0.001[b] |
| Female (n = 194) | Endpoint archive | 58 | 92.1 | 70 | 88.6 | 50 | 100.0 | 0.046 |
| | Mean ± SD | 272.81 ± 61.26 | | 308.23 ± 238.75 | | 235.10 ± 52.43 | | 0.036[b] |
| | Mean difference | −253.34 ± 59.96 | | −213.45 ± 203.77 | | −300.69 ± 54.60 | | 0.007[b] |
| T2DM (n = 190) | Endpoint archive | 64 | 92.8 | 54 | 90.0 | 91 | 100.0 | 0.051 |
| | Mean ± SD | 277.00 ± 67.68 | | 322.75 ± 271.69 | | 238.93 ± 51.57 | | 0.018[b] |
| | Mean difference | −240.85 ± 62.52 | | −200.75 ± 263.73 | | −304.18 ± 52.71 | | 0.001[b] |
| Dyslipidemia (n = 334) | Endpoint archive | 118 | 95.6 | 109 | 91.6 | 92 | 100.0 | 0.013 |
| | Mean ± SD | 267.16 ± 61.55 | | 303.21 ± 197.72 | | 236.41 ± 47.73 | | 0.001[b] |
| | Mean difference | −249.59 ± 61.65 | | −217.44 ± 190.67 | | −303.76 ± 50.88 | | <0.001[b] |
| Tertile 3 (n = 183) | | | | | | | | |
| Male (n = 90) | Endpoint archive | 13 | 76.5 | 27 | 93.1 | 43 | 97.7 | 0.021 |
| | Mean ± SD | 374.00 ± 61.28 | | 350.21 ± 53.98 | | 305.66 ± 47.93 | | <0.001[b] |
| | Mean difference | −284.47 ± 70.70 | | −291.17 ± 57.84 | | −356.84 ± 51.87 | | <0.001[b] |
| Female (n = 93) | Endpoint archive | 13 | 56.5 | 10 | 47.6 | 46 | 93.9 | <0.001 |
| | Mean ± SD | 359.04 ± 43.51 | | 372.76 ± 52.53 | | 306.84 ± 45.86 | | <0.001[b] |
| | Mean difference | −299.13 ± 57.45 | | −286.09 ± 57.26 | | −352.61 ± 45.75 | | <0.001[b] |
| T2DM (n = 91) | Endpoint archive | 10 | 66.7 | 15 | 71.4 | 51 | 92.7 | 0.013 |
| | Mean ± SD | 360.07 ± 66.49 | | 366.14 ± 63.91 | | 313.44 ± 46.76 | | <0.001[b] |
| | Mean difference | −307.86 ± 64.72 | | −281.47 ± 55.51 | | −326.61 ± 60.63 | | <0.001[b] |
| Dyslipidemia (n = 159) | Endpoint archive | 23 | 69.7 | 32 | 74.4 | 79 | 95.2 | <0.001 |
| | Mean ± SD | 361.52 ± 52.31 | | 356.72 ± 55.80 | | 307.81 ± 46.93 | | <0.001[b] |
| | Mean difference | −295.18 ± 65.06 | | −298.21 ± 59.26 | | −353.41 ± 49.48 | | <0.001[b] |

Notes:
SD, standard deviation.
[a] Chi-squared test.
[b] One-way ANOVA test.

−225.75 ± 78.53 µmol/L and −273.02 ± 204.54 µmol/L to −308.93 ± 72.97 µmol/L in group A, B, and C respectively (Table 3). The tertile subgroup of uric acid levels also indicated the optimal treatment outcomes in group C and clearest in tertile 3 which was the same as total population analysis (Table 3).

**Table 4  Multivariable analysis assessed related factors to uric acid treatment outcomes.**

| Related factors | Endpoint archive proportion (%) | Absolute difference in proportion (%) | OR | 95% CI | p |
|---|---|---|---|---|---|
| Aged > 60 years | 92.5 | 3.7 | 0.55 | [0.23–1.33] | 0.190 |
| Male | 97.4 | 7.6 | 4.41 | [0.74–26.05] | 0.101 |
| BMI ≥ 25 (kg/m$^2$) | 97.4 | 4.4 | 2.49 | [0.55–11.16] | 0.233 |
| NYHA ≥ 3 | 85.7 | 7.8 | 0.41 | [0.13–1.31] | 0.132 |
| EF ≤ 40 (%) | 94.9 | 5.6 | 2.25 | [1.19–4.25] | 0.013 |
| Hypertension | 93.4 | 0.5 | 1.11 | [0.39–3.11] | 0.843 |
| T2DM | 92.4 | 2.0 | 0.73 | [0.39–1.38] | 0.335 |
| Dyslipidemia | 93.8 | 3.7 | 1.50 | [0.55–4.11] | 0.428 |
| History of CAD | 94.3 | 1.7 | 0.98 | [0.50–1.94] | 0.964 |
| History of CI | 95.3 | 2.8 | 1.71 | [0.81–3.58] | 0.156 |
| Smoking | 97.0 | 5.6 | 0.51 | [0.09–2.90] | 0.447 |
| Alcohol consumption | 97.6 | 7.0 | 2.14 | [0.51–8.97] | 0.299 |

**Note:**

CAD, coronary artery disease; 95% CI, 95% of confidence interval; CI, cerebral infarction; EF, ejection fraction; NYHA, The New York Heart Association; OR, odd ratio.

### Related factors affecting uric acid treatment outcome

Multivariable analysis indicated that EF ≤ 40% related to the endpoint archive proportion of the study population, with the OR as 2.25, 95% CI [1.19–4.25], and $p = 0.013$ (Table 4). Males, BMI ≥ 25 (kg/m$^2$), hypertension, dyslipidemia, history of CI, and alcohol consumption also have the OR > 1 but were not statistically significant (Table 4).

## DISCUSSION

### Principal findings

Our study evaluated the effectiveness of serum uric acid lowering treatment of SGLT2 inhibitors, and ARB in HF patients. The study outcome clearly described the optimal therapy for reducing uric acid levels in HF patients as the combination of losartan and dapagliflozin, and the effectiveness did not change through variable population groups such as sex, T2DM, and dyslipidemia. In addition, the multivariable analysis indicated the EF < 40% related to the achievement proportion of uric acid outcome. Therefore, our study could be used as a strong medical reference for reducing uric acid level drug selection in HF patients, especially in the Vietnamese population.

### Possible explanations and comparison with other studies

The overall baseline characteristics of our study indicated a small difference between the three groups. Our study population has an age ranging from 67.34 to 68.50 years and was balanced in the male and female ratio. Consistent with other studies such as *Doehner et al. (2022)*, *McMurray et al. (2019)*, *Matsumura et al. (2015)*, *Ferreira et al. (2023)* the age was around 65–70 years, and balance in male and female (*Matsumura et al., 2015*; *Wang et al., 2023b*). The mean concentration of uric acid at baseline of groups A, B, and C was 495.41 ± 109.39 μmol/L, 504.56 ± 111.80 μmol/L, and 548.78 ± 123.49 μmol/L,

respectively. The baseline concentration of uric acid was consistent (*Doehner et al., 2022*; *Ferreira et al., 2023*) and different with some study (*Matsumura et al., 2015*; *McMurray et al., 2019*; *Wang et al., 2023b*). The difference was due to the difference in regional and race populations.

The efficacy of dapagliflozin in reducing uric acid levels as high as the total population and indicated in sub-groups of the population which proves the effect did not change between different demographics such as sex, comorbidities like T2DM and dyslipidemia (Tables 2 and 3). The endpoint archive proportion was 90.3% and the mean difference was −217.00 ± 146.17 (Table 2). However, in tertile 3 of the uric acid levels, dapagliflozin monotherapy showed a low effect in reducing uric acid concentration. Consistent with the study of *Doehner et al. (2022)*, which investigated the efficacy of empagliflozin in 3,676 HF patients, 4 weeks of follow-up showed the mean reduction of uric acid compaired with control group as −1.12 ± 0.04 mg/dL, $p < 0.0001$ and remained lower throughout follow-up, with a similar reduction in all prespecified subgroups (*Doehner et al., 2022*). A study by *Wang et al. (2023b)* in the T2DM population also showed a high effect of uric acid reduction compared with the control group. Therefore, SGLT2i especially dapagliflozin could be a reducing uric acid agent in light hyperuricemia patients and reduce the effect as higher uric acid concentration groups.

Assessed the uric acid reduction effect of losartan in group A of treatment indicated that the achievement endpoint proportion was 92.3%, the mean difference in the total population was −229.62 ± 76.65 μmol/L (Table 2), and the affected did not change through difference demographic of the population (Table 3). The affected reduced in tertile 3 of the uric acid concentration, which indicated as low affected at higher concentrations of serum uric acid. Our outcome was similar to other studies such as *Matsumura et al. (2015)* assessed the uric acid reduction effect of losartan compared with the control group as the mean difference was 0.53 mg/dL ($p = 0.01$). *Ferreira et al. (2023)* study assessed the efficacy of losartan in HF patients through the tertile 3 of uric acid concentration also indicated the reduction of effect in the higher group of uric acid concentration. Therefore, ARBs as same as SG2T2i would be a reducing uric acid agent in light hyperuricemia patients and reduce the effect as higher uric acid concentration groups.

Our study aimed to assess the optimal outcome between the three groups. The combination of losartan and dapagliflozin indicated the optimal regimen. The achievement endpoint proportion and mean difference after 8 weeks of follow-up did not change through tertile of uric acid concentration (Table 2). In addition, the sub-group of population analysis showed the same effect of uric acid reduction between groups (Table 3). There were very few studies that have assessed the combination of losartan and dapagliflozin like our study. Therefore, the combination of ARBs and SGLT2i in HF treatment especially losartan and dapagliflozin indicated the optimal regimen in the HF population. In addition, our study also assessed the related factors to the treatment outcome. The multivariable analysis indicated the reduction of EF < 40% was related to the reduction of endpoint reaching proportion (Table 4).

### Strengths and weaknesses of the study

Our study had a sample collection process with clearly designed including and excluding criteria. All study participants volunteered and benefited from the study. Study methods were clearly described and reproducible. Our sample size was big enough for representative of the study population and as strong medical evidence. The study outcome clearly showed the differential effect between the three treatment groups through 8 weeks of follow-up. Therefore, our study outcomes could be used as medical evidence to choose a treatment regimen and ensure the safety of using the drug on patients.

However, our study was evaluated only in 1 hospital, it might lead to bias in baseline characteristics. Therefore, a multicenter study with the same or larger sample size is required for a better representation of the study population. Additionally, our study did not assess the combination of medications used and the cause of HF, which confounding factors could affect the study outcomes. In our study with a follow-up period of 8 weeks, patients used the drug longer than the evaluation period. Therefore, a study with a longer evaluation period is needed to give stronger evidence for the efficacy and safety effects of the three regimens. Furthermore, in the era of ARNI in HF therapy, the additional use of losartan to reduce uric acid levels in HF therapy may lead to potential impacts. The future outlook for the betterment of the uric acid reduction therapy application required an appropriate study design to assess the potential impacts and confounding factors surrounding the usage of losartan and dapagliflozin in monotherapy and combinations.

## CONCLUSIONS

Losartan and dapagliflozin used as monotherapy or combination therapy indicated a high proportion of endpoint achivement. However, the optimal therapy for reducing uric acid levels in HF patients was the combination of losartan and dapagliflozin, and the effectiveness did not change through sex, T2DM, and dyslipidemia patients. Additionally, EF < 40% indicated a relationship with higher endpoint achievement. Clinical physicians could use our study outcome as a reference for drug selection in reducing uric acid in HF patients in Vietnam.

## ACKNOWLEDGEMENTS

We would like to thank Can Tho University of Medicine and Pharmacy for creating favorable conditions for this study to be carried out.

### Funding

The authors received no funding for this work.

### Competing Interests

The authors declare that they have no competing interests.

## Author Contributions

- Tuong Le Trong Huynh conceived and designed the experiments, performed the experiments, prepared figures and/or tables, authored or reviewed drafts of the article, and approved the final draft.
- Phong Thanh Pham conceived and designed the experiments, performed the experiments, prepared figures and/or tables, authored or reviewed drafts of the article, and approved the final draft.
- Hien Dieu Tran conceived and designed the experiments, performed the experiments, prepared figures and/or tables, authored or reviewed drafts of the article, and approved the final draft.
- Nhan Dinh Tran conceived and designed the experiments, performed the experiments, prepared figures and/or tables, authored or reviewed drafts of the article, and approved the final draft.
- Duong Van Tran conceived and designed the experiments, performed the experiments, prepared figures and/or tables, authored or reviewed drafts of the article, and approved the final draft.
- Bao Lam Thai Tran analyzed the data, prepared figures and/or tables, authored or reviewed drafts of the article, and approved the final draft.
- Khoa Dang Dang Tran analyzed the data, prepared figures and/or tables, authored or reviewed drafts of the article, and approved the final draft.
- Toan Hoang Ngo analyzed the data, prepared figures and/or tables, authored or reviewed drafts of the article, and approved the final draft.
- Son Kim Tran analyzed the data, prepared figures and/or tables, authored or reviewed drafts of the article, and approved the final draft.

## Human Ethics

The following information was supplied relating to ethical approvals (*i.e.*, approving body and any reference numbers):

The study was approved by the Ethics Committee in Biomedical Research of Can Tho University of Medicine and Pharmacy with Decision No. 182/HDDD-DHYDCT at December 22, 2022.

## Data Availability

The data is available in the Supplemental Files and figshare: Hoang Ngo, Toan (2024). RawData.xlsx. figshare. Dataset. https://doi.org/10.6084/m9.figshare.27291606.v1.

## Supplemental Information

Supplemental information for this article can be found online at http://dx.doi.org/10.7717/peerj.18595#supplemental-information.

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
