# Peer review of "Losartan and dapagliflozin combination therapy in reducing uric acid level compared to monotherapy in patients with heart failure"

_PeerJ, doi:10.7717/peerj.18595_

## Round 0.1 · original submission · Major Revisions

There are several aspects to be considered, as indicated by reviewers. Note that acceptance after the review cannot be guaranteed.

Reviewer 1 ·

Basic reporting

The article is well written.
HF coexists with CKD, and background information should include that. In this context, the levels of uric acid will be higher.

Experimental design

The methodology has to be clear about how CKD was defined. Was it based on GFR or absolute creatinine values? What was the cutoff? Hyperuricemia is prevalent even in stage 1 CKD.

The dosage of diuretics in HF is an important confounding factor. How was this accounted for? Reducing the dosage of diuretics can lower uric acid levels.

Validity of the findings

The efficacy of therapy also depends on how well these patients were treated with GDMT for HF. Either mentioning that all were stable HF patients or describing the average dose of beta-blockers and mineral receptor antagonists would have added more value. The uric acid levels of HF patients on MRA may also be discussed.

Additional comments

Nil

Reviewer 2 ·

Basic reporting

A very nice idea for a manuscript. The language and grammar should be revised thoroughly.

Experimental design

After the 2021 ESC guidelines for heart failure patients should receive the 4 pillars of HFrEF (ACEi/ARNI, SGLT2i, MRAs, beta blockers). Having less than that should be explained? ARBs have been downgraded in the treatment algorithm to be only used when ACEi or ARNI are not possible.

Validity of the findings

The tables don not show the left ventricular dimensions. There is no mention of the concomitant medications.

Reviewer 3 ·

Basic reporting

I received for review an original research article entitled "Losartan and dapagliflozin combination therapy in reducing uric acid level compared to monotherapy in patients with heart failure", prepared by Tuong Le Trong Huynh, which was submitted to the PeerJ. Cardiovascular diseases are one of the most serious health problems of many modern societies, especially in developed countries. Heart failure is a particularly important problem, because it is the final stage of many heart diseases. The population of patients with heart failure is heterogeneous. However, there is no doubt that heart failure is a serious clinical condition that significantly affects prognosis. It is therefore important to conduct research on this topic, which may contribute to better control of metabolic disorders such as hyperuricemia in the population of people with heart failure. In my opinion, the manuscript is quite well prepared and represents quite a high substantive and scientific value. However, I would like to suggest some modifications that, in my opinion, are necessary to further improve the quality and attractiveness of the presented manuscript.
1) I think the introduction is a bit too general and laconic. It is worth briefly mentioning the heterogeneity of population with heart failure (division by left ventricular systolic function, division by dominant symptomatology), as well as the most important elements of the pathogenesis of heart failure. It is worth emphasizing why the topic of hyperuricemia is important. It is worth mentioning the metabolic syndrome and the fact that hyperuricemia is a typical abnormality found in patients with metabolic syndrome, although it is not one of its diagnostic criteria. Metabolic syndrome, on the other hand, predisposes to the development of atherosclerotic cardiovascular disease. (https://doi.org/10.3390/medicina60071080)
2) What was the etiology of heart failure in the study population? Was the entire population diagnosed with heart failure of the same etiology? If not, did the authors attempt to perform subgroup analyses according to etiology?
3) When describing the strengths and weaknesses of the study, it is worth noting what, in the authors’ opinion, is a strength and what is a weakness.
4) The conclusions is too short and superficial. It is worth expanding on it.

Experimental design

No comment.

Validity of the findings

No comment.

Additional comments

No comment.

---

## Round 0.2 · Minor Revisions

Please, note that there are some points to be considered before the manuscript can be accepted (particularly the points raised by reviewer 1).

Reviewer 1 ·

Basic reporting

No comment

Experimental design

No comment

Validity of the findings

After the revision, the conclusions drawn and conveyed are a bit confusing. The study should clearly convey the message of losartan reducing uric acid in HF and its impact on clinical practice.

Additional comments

In the present era of the benefits of ARNI therapy in HF, losartan is only a second option in resource-challenged situations. How uric acid reduction with losartan can impact HF therapy is debatable. Also, the confounding factors leading to hyperuricemia in HF need to be addressed. This may be discussed in the text.

Reviewer 2 ·

Basic reporting

The manuscript is simple and clear.

Experimental design

The design is well explained and the limitations including the absence of information on the use of combination therapy in heart failure is mentioned.

Validity of the findings

The findings are valid taking into consideration the absence of data on other medications.

Reviewer 3 ·

Basic reporting

I received for review a revised version of the original research article entitled "Losartan and dapagliflozin combination therapy in reducing uric acid level compared to monotherapy in patients with heart failure", prepared by Tuong Le Trong Huynh, which was submitted to the PeerJ. In my opinion, the paper has been significantly improved. I have no further comments.

Experimental design

I have no further comments.

Validity of the findings

In my opinion, the paper represents high scientific value.

Additional comments

I recommend the paper for publication in its current form.

---

## Round 0.3 · accepted · Accept

Thank you for correcting the text according to the reviewers indications.